# The Amino Acid-mTORC1 Pathway Mediates APEC TW-XM-Induced Inflammation in bEnd.3 Cells

**DOI:** 10.3390/ijms22179245

**Published:** 2021-08-26

**Authors:** Dong Zhang, Shu Xu, Yiting Wang, Peng Bin, Guoqiang Zhu

**Affiliations:** 1College of Veterinary Medicine, Yangzhou University, Yangzhou 225009, China; zdd8911@163.com (D.Z.); MX120170707@yzu.edu.cn (S.X.); MX120180700@yzu.edu.cn (Y.W.); binpeng15@mails.ucas.ac.cn (P.B.); 2Jiangsu Co-Innovation Center for Prevention and Control of Important Animal Infectious Diseases and Zoonoses, Yangzhou 225009, China; 3Joint International Research Laboratory of Agriculture and Agri-Product Safety, The Ministry of Education of China, Yangzhou University, Yangzhou 225009, China

**Keywords:** APEC XM, amino acids, amino acid transporter, blood–brain barrier, mTORC1 pathway, inflammation

## Abstract

The blood–brain barrier (BBB) is key to establishing and maintaining homeostasis in the central nervous system (CNS); meningitis bacterial infection can disrupt the integrity of BBB by inducing an inflammatory response. The changes in the cerebral uptake of amino acids may contribute to inflammatory response during infection and were accompanied by high expression of amino acid transporters leading to increased amino acid uptake. However, it is unclear whether amino acid uptake is changed and how to affect inflammatory responses in mouse brain microvascular endothelial (bEnd.3) cells in response to Avian Pathogenic *Escherichia coli* TW-XM (APEC XM) infection. Here, we firstly found that APEC XM infection could induce serine (Ser) and glutamate (Glu) transport from extracellular into intracellular in bEnd.3 cells. Meanwhile, we also shown that the expression sodium-dependent neutral amino acid transporter 2 (SNAT2) for Ser and excitatory amino acid transporter 4 (EAAT4) for Glu was also significantly elevated during infection. Then, in amino acid deficiency or supplementation medium, we found that Ser or Glu transport were involving in increasing SNAT2 or EAAT4 expression, mTORC1 (mechanistic target of rapamycin complex 1) activation and inflammation, respectively. Of note, Ser or Glu transport were inhibited after SNAT2 silencing or EAAT4 silencing, resulting in inhibition of mTORC1 pathway activation, and inflammation compared with the APEC XM infection group. Moreover, pEGFP-SNAT2 overexpression and pEGFP-EAAT4 overexpression in bEnd.3 cells all could promote amino acid uptake, activation of the mTORC1 pathway and inflammation during infection. We further found mTORC1 silencing could inhibit inflammation, the expression of SNAT2 and EAAT4, and amino acid uptake. Taken together, our results demonstrated that APEC TW-XM infection can induce Ser or Glu uptake depending on amino acid transporters transportation, and then activate amino acid-mTORC1 pathway to induce inflammation in bEnd.3 cells.

## 1. Introduction

Bacterial meningitis is a serious brain disease that is associated with high morbidity and mortality [1,2]. Many people who survive meningitis-related infections develop permanent nerve injuries resulting from a lack of timely treatment [3,4,5]. Among meningitis-causing *Escherichia coli*, avian pathogenic *E. coli* (APEC) strains and neonatal meningitis *E. coli* (NMEC) have highly homologous virulence factors and show similar pathogenic mechanisms [6,7,8]. APEC is thought to be a potentially zoonotic pathogen that not only negatively affects the poultry industry but also poses a risk to human health [9,10]. Prominent pathophysiological characteristics of bacterial meningitis are the destruction of the integrity of the blood–brain barrier (BBB). Brain microvascular endothelial cells (BMECs) are the main components of the BBB and are widely used as a model to study the BBB in vitro. Although BMECs normally regulate the transportation and absorption of nutrients to maintain the integrity of the BBB [11], they can produce a large number of pro-inflammatory factors when infected by meningitis bacteria, thereby enhancing the disruption of BBB integrity [12,13]. Tumor necrosis factor (TNF)-α is associated with increased BBB permeability in infant rats [12]. In addition, inducible nitric oxide synthase (iNOS) was upregulated in endothelial cells resulting in disruption of the integrity of the BBB during infection [14]. Therefore, a better understanding of the processes and mechanisms underlying BBB inflammation may contribute to the development of novel therapeutics against bacterial meningitis.

Amino acids (AAs) are the synthetic precursors of proteins and have a crucial role in regulating organismal metabolism [15], maintaining redox homeostasis in infectious diseases [16,17,18]. Pathogens can induce changes in AAs uptake, which is involved in inducing a significant inflammatory reaction and neuropathology [16]. For example, serine deprivation weakened the expression of pro-inflammatory factor IL-1β (interleukin-1beta) in macrophages [19]. In bacterial meningitis, excitatory amino acids (glutamate and aspartate) were significantly elevated in the brain after meningitis *E. coli* K1 infection, and play a critical role in the pathogenesis of neuronal cell damage [20]. Regarding amino acid uptake, transmembrane transporters (solute carriers, SLCs) are usually responsible for maintaining intracellular homeostasis. Among the SLCs, AA transporters facilitate the exchange or uptake of AAs. Meanwhile, the expressions of amino acid transporters are uncovered that are involved in immune responses under different conditions, including aging, infection, and cancer [21,22,23,24,25]. The mechanistic target of rapamycin (mTOR) as conserved serine/threonine protein kinase includes two functionally distinct complexes named mTOR complex1 (mTORC1) and mTORC2, which are involved in mediating immune responses to defense pathogens. mTORC1 promotes mRNA translation via phosphorylation of its effectors S6K1 (ribosomal protein S6 Kinase 1) and 4E-BP1(eukaryotic initiation factor 4E binding protein 1). mTORC2 is responsible for the regulation of AKT. Usually, amino acids, energy deficit, and hypoxia could activate the mTORC1 pathway, and insulin/insulin growth factor (Ins/IGF) is responsible for activating the mTORC2 pathway. In bacterial infection reports, it has been shown that mTORC1 was associated with inducing inflammation, and promoted bacterial infection [26,27]. Additionally, AAs transporters could activate the mTORC1 pathway for inducing inflammation to maintain the integrity of the BBB by regulating intracellular amino acids and may be new promising therapeutic targets [28,29]. However, whether AAs play a role during *E. coli* meningitis-related infection in the BBB remains unknown.

In this study, we explored the relationship between the levels of AAs and the production of pro-inflammatory factors in bEnd.3 cells during APEC TW-XM (avian pathogenic Escherichia coli TW-XM) infection. We found that APEC XM infection-induced inflammation was associated with the intracellular increased serine (Ser) and glutamate (Glu) levels. Notably, APEC XM could induce high expression of SNAT2 and EAAT4 which transported Ser and Glu, respectively, and activate the mTORC1 pathway in bEnd.3 cells. Thus, whether Ser and Glu depend on transporters and are involved in APEC TW-XM-induced inflammation by activating the mTORC1 pathway in bEnd.3 cells will be elucidated in this work.

## 2. Results

### 2.1. APEC XM Induced Changes of Serine and Glutamate Uptake in bEnd.3 Cells

Several studies have shown that pathogenic bacterial infections can affect amino acid homeostasis, resulting in inflammation in T cells [30,31,32]. Therefore, we measured the intracellular and extracellular (supernatant) 18 AA levels in APEC XM-infected bEnd.3 cells at 1 h and 12 h post-infection. Compared with uninfected controls, the intracellular levels of 16 AAs, including Ser, alanine (Ala), Glu, and aspartate (Asp), were higher in the infected cells at 1 h post-infection (Figure 1A, Appendix A). Interestingly, the levels of Ser, Ala, Glu, and Asp were significantly decreased in the supernatants (Figure 1B), whereas no significant change was observed in the uptake of the other 14 AAs (Appendix A). At 12 h post-infection, the intracellular concentrations of 14 AAs were significantly increased in infected cells (Appendix A); however, no significant differences were observed in the supernatant concentrations of these amino acids, except for Glu (Appendix A). These results suggested that the increased intracellular levels of Ser, Ala, Glu, and Asp may depend on amino acid transporters at 1 h post-infection. However, there was no significant change in the supernatant at 12 h post-infection, which increased intracellular amino acids, perhaps not mainly caused by amino acid transporters. In addition, the DMEM component does not contain alanine and aspartic acid, and there may be other factors associated with amino acid change during infection. Hence, we mainly focused on exploring whether Ser and Glu levels’ change depends on amino acid transporters, resulting in activation of the mTORC1 pathway and inducing an inflammatory response at 1 h post-infection.

### 2.2. APEC XM Infection Increases the Expression of SNAT2 and EAAT4 in bEnd.3 Cells

Amino acid uptake mainly depends on amino acid transporters, and amino acid transporters are responsible for maintaining host AA metabolism balance or defending against pathogen infection by regulating AA levels [17,33,34,35]. To determine whether the changes of Ser and Glu uptake are mediated through amino acid transporters in bEnd.3 cells at 1-h post-APEC XM infection, we measured the expression levels of Ser, Glu transporters. At the mRNA level, the expression of slc38a2 (SNAT2, sodium-dependent neutral amino acid transporter 2) coding for transporters common to Ser increased significantly at 1 h post-infection (Figure 2A), while the mRNA expression levels of the other Ser transporters (slc1a4, slc1a5, slc6a14, slc7a9, slc7a10, slc38a1, and slc38a4) showed no significant differences (Figure 2A). The expression levels of the transporters slc1a6 (EAAT4, excitatory amino acid transporter 4), shared by Glu, were significantly increased 1 h after infection (Figure 2A); however, the mRNA expression levels of *slc1a2* and *slc1a7* were significantly reduced difference and others showed no significant difference at 1 h post-infection (Figure 2A).

We next measured the abundance of SNAT2 and EAAT4 by Western blot. The SNAT2 and EAAT4 protein levels showed significantly increased levels of infected bEnd.3 cells when compared with uninfected control (Figure 2B). These results suggested that the expression levels of amino acid transporters are markedly upregulated during the early stages of APEC XM infection.

### 2.3. APEC XM Activates the mTORC1 Pathway and Inflammatory Responses in bEnd.3 Cells

The mTORC1 pathway can mediate amino acid-related effects, including regulation of the production of pro-inflammatory factors [36]. To determine whether APEC XM infection activates the mTORC1 pathway and promotes the production of pro-inflammatory factors, we starved bEnd.3 cells for 6 h in a serum-free medium, and then infected the cells with APEC XM for 1 h [37,38]. The results showed that the mTORC1 pathway was significantly activated by APEC XM infection, as evidenced by the results of p-S6K and p-4E-BP1staining (Figure 3A). 

In addition, numerous studies have shown that meningitis *E. coli* induces a severe inflammatory response in the BBB [39,40]. Here, we showed that APEC XM infection upregulated the expression of pro-inflammatory factors in bEnd.3 cells, including that of IL-6, IL-1β, TNF-α, and iNOS (Figure 3B). These results indicated that APEC XM infection activates the mTORC1 pathway and induces inflammatory responses in bEnd.3 cells.

### 2.4. Ser Involved in APEC XM-Induced the Expression of SNAT2, Activation of mTOR Pathway and Inflammatory Responses in bEnd.3 Cells

Then, we explored whether the change of Ser transport from extracellular to intracellular promotes increased expression of SNAT2 during APEC XM infection, and is further involved in activating the mTOR pathway and inducing inflammatory responses. We infected the bEnd.3 cells with APEC XM in the 1 mM Ser supplement or the Ser deficiency medium for 1 h. In the Ser deficiency group, the *slc38a2* gene and SNAT2 expression were significantly reduced when compared with the APEC XM infection group as well as the 1 mM Ser supplement group (Figure 4A). Meanwhile, the intracellular Ser level in the Ser deficiency group showed no significant change compared with the control group during APEC XM infection (Figure 4B). In the 1 mM Ser supplement group, the intracellular Ser level was significantly increased compared with the control group after APEC XM infection. These results suggest that the increase of intracellular Ser is mainly caused by SNAT2 transport, which transports extracellular Ser into intracellular during infection.

Since bEnd.3 cells infected with APEC XM showed increased expression of SNAT2, we investigated whether increased expression of SNAT2 allows Ser to activate the mTORC1 pathway (mTOR, p-mTOR, p-4E-BP1 and p-S6K). The mTORC1 pathway in APEC XM-infected bEnd.3 cells were activated with Ser and inactivated in the Ser deficiency medium during APEC XM infection (Figure 4C). Notably, Ser supplementation significantly increased the expression of IL-6, IL-1β, and TNF-α in bEnd.3 cells compared with the control group and Ser deficiency group. Simultaneously, the expression of these pro-inflammatory factors in the 1 mM Ser supplement group was significantly higher than those in the APEC XM infection group. However, the expression of iNOS was not significantly increased between the 1 mM Ser supplement medium and the Ser deficiency medium (Figure 4D). Taken together, the data demonstrated that Ser promoted increased expression of SNAT2 mediated by APEC XM infection, and Ser transported into intracellular involved in activating the mTOR pathway and inducing inflammatory responses.

### 2.5. Glu Involved in APEC XM-Induced the Expression of EAAT4, Activation of mTOR Pathway and Inflammatory Responses in bEnd.3 Cells

Similarly, we also explored whether the change of Glu in extracellular and intracellular promotes the increased expression of EAAT4 during APEC XM infection, and then further investigated the effect of increased intracellular Glu on activating the mTOR pathway and inducing inflammatory responses. In the Glu deficiency group, the *slc1a6* gene and EAAT4 expression were significantly reduced when compared with the APEC XM infection group as well as the 1 mM Glu supplement group (Figure 5A). Meanwhile, the intracellular Glu level in the Glu deficiency group showed no significant change compared with the control group during APEC XM infection (Figure 5B). In the Glu deficiency group, the intracellular Glu level was significantly decreased compared with the APEC XM infection group, as well as the 1 mM Glu supplement group. These data suggest that APEC XM infection can contribute to transporting extracellular Glu into intracellular by increasing the expression of EAAT4.

Next, we investigated whether Glu promotes activation of the mTORC1 pathway and induces inflammation. The mTORC1 pathway, including mTOR, p-mTOR, p-4E-BP1, p-S6K, were activated with Glu in APEC XM-infected bEnd.3 cells, and inactivated in the Glu deficiency medium during APEC XM infection (Figure 5C). During infection, the expression of IL-1β, IL-6, TNF-α, and iNOS in the Glu deficiency group was significantly decreased compared with the APEC XM infection group. Simultaneously, the expression of these pro-inflammatory factors in the 1 mM Glu supplement group was significantly higher than those in the APEC XM infection group. However, the expression of TNF-α and iNOS was not significantly increased in the 1 mM Glu supplement medium (Figure 5D). The results suggest that the transportation of Glu from extracellular into intracellular depends on the increased expression of EAAT4 induced by APEC XM, and the elevated intracellular Glu participated in the activation of the mTOR pathway and inflammatory responses.

### 2.6. SNAT2 Promotes Ser Uptake to Activate the mTORC1 Pathway and Inflammatory Responses during APEC XM-Infection

Based on the observed changes in the levels of amino acid uptake, activation of the mTORC1 pathway, and increased expression of pro-inflammatory factors following APEC XM infection, we hypothesized that transport-mediated cellular uptake of amino acids could lead to the activation of the mTORC1 pathway and subsequent induction of pro-inflammatory factors. To test this hypothesis, bEnd.3 cells were transfected with SNAT2 siRNA or a pEGFP-SNAT2 before infection. After APEC XM infection, cells were detected. At the RNA and protein levels, SNAT2 knockdown significantly inhibited the expression of SNAT2, and 1 mM Ser also lost the ability to increase the expression of SNAT2 during infection (Figure 6A). APEC XM can induce marked SNAT2 overexpression in the pEGFP+SNAT2 group, whereas the expression of SNAT2 was significantly reduced in the pEGFP+SNAT2+Ser deficiency group (Figure 6A). These data suggest that SNAT2 was involved in Ser transport and Ser was important to promote the expression of SNAT2 during APEC XM infection.

In the Ser test, when SNAT2 was silenced, the intracellular Ser content was significantly reduced, even with the supplementation of 1 mM Ser, the intracellular Ser content was still significantly lower than that of the APEC XM infection group (Figure 6B), and the corresponding extracellular content of Ser was significantly more than that of the APEC XM infection group (Figure 6B), suggesting that Ser transport depended on SNAT2. Meanwhile, overexpression of SNAT2 also promoted the transportation of Ser from extracellular into intracellular during APEC XM infection (Figure 6B). 

To explore the possible mechanisms of SNAT2-transport Ser to promote APEC XM infection in bEnd.3 cells, activation of the mTORC1 pathway, and associated pro-inflammatory (IL-6, IL-1β, TNF-α, and iNOS) were examined. In the complete medium, we found that the expression of p-mTOR, p-4E-BP1 and p-S6K was significantly activated in the APEC XM infection group and the pEGFP-SNAT2 group, compared with the SNAT2 silence group (Figure 6C). SNAT2 silence resulted in no significant activation of the mTORC1 pathway even in a medium supplemented with 1 mM Ser, yet there was no marked difference in the expression of mTOR (Figure 6C). Similarly, SNAT2 overexpression markedly increased the mRNA levels of pro-inflammatory factors IL-6, IL-1β, TNF-α, and iNOS (Figure 6D), and these pro-inflammatory factors were significantly suppressed when SNAT2 was silenced (Figure 6D). These data indicate that SNAT2 could be involved in inducing the activation of mTOR and the expression of pro-inflammatory factors by transporting Ser.

### 2.7. EAAT4 Promotes Glu Transport to Activate mTORC1 and Inflammatory Responses during APEC XM Infection

Similarly, to study the function of EAAT4 in the APEC XM-infected bEnd.3 cells, we constructed SNAT2 siRNA or a pEGFP-SNAT2 and then transfected it into bEnd.3 cells before infection. After APEC XM infection, cells were detected. At the RNA and protein levels, EAAT4 silence significantly inhibited the expression of EAAT4 even in the medium supplemented with 1 mM Glu (Figure 7A). APEC XM can induce marked EAAT4 overexpression in the pEGFP+EAAT4 group, whereas the expression of EAAT4 was significantly reduced in the pEGFP+EAAT4+Glu deficiency group (Figure 7A). In the Glu test, when EAAT4 was silenced, the intracellular Glu content was significantly reduced, even with the supplementation of 1 mM Glu, the intracellular Glu content was still significantly lower than that of the APEC XM infection group (Figure 7B), and the corresponding extracellular content of Glu was significantly greater than that of the APEC XM infection group (Figure 7B), suggesting that Glu transport depended on EAAT4. Meanwhile, overexpression of EAAT4 was shown to mainly promote the transport of Glu from extracellular into intracellular during APEC XM infection (Figure 7B). 

To explore the possible mechanisms of EAAT4-transport Glu to promote APEC XM infection in bEnd.3 cells, activation of the mTORC1 pathway, and its associated pro-inflammatory factors expression (IL-6, IL-1β, TNF-α, and iNOS), were examined. In the complete medium, we found that the mTORC1 pathway was significantly activated in the APEC XM infection group and the pEGFP-EAAT4 group (Figure 7C). However, EAAT4 silence resulted in no significant activation of the mTORC1 pathway even in the medium supplemented with 1 mM Glu (Figure 7C). Similarly, EAAT4 overexpression markedly increased the mRNA levels of pro-inflammatory factors IL-6, IL-1β, TNF-α, and iNOS (Figure 7D), and these pro-inflammatory factors were significantly suppressed when EAAT4 was silenced (Figure 7D). These data indicated EAAT4 could be involved in inducing the activation of mTOR and the expression of pro-inflammatory factors by transporting Glu.

### 2.8. APEC XM Induces Amino Acid-Driven Inflammation through The mTORC1 Pathway

The mTORC1 pathway is activated by AAs and regulates inflammation [17,18,36]. We next investigated whether the mTORC1 pathway affects Ser and Glu uptake, the expression of SNAT2 and EAAT4, as well as the expression of pro-inflammatory factors. As expected, after silencing the mTOR gene, APEC XM-induced activation of the mTORC1 pathway was inhibited, and activation of mTORC1 pathway was still suppressed in the 1 mM Ser supplement and the 1 mM Glu supplement mediums (Figure 8A). Additionally, the expression of IL-6, IL-1β, TNF-α, and iNOS was significantly decreased when mTOR was silent (Figure 8D). Moreover, the RNA and protein levels of SNAT2 and EAAT4 were also markedly decreased after mTOR silence (Figure 8B). Therefore, Ser and Glu transport from extracellular into intracellular was inhibited (Figure 8C). Collectively, during APEC XM infection, the results implied that the mTORC1 pathway was involved in regulating the expression of pro-inflammatory factors, and also affected the expression of SNAT2 and EAAT4 resulting in the suppression of Ser and Glu from extracellular into intracellular.

## 3. Discussion

Due to amino acids’ involvement in the interaction between bacteria and host, Ser and Glu have always been the focus of research [17,41]. Ser and Glu have increasingly been involved in promoting central nervous system diseases, such as Alzheimer’s disease (AD), Parkinson’s disease (PD), and multiple sclerosis [42,43,44,45,46,47]. However, the function of Ser and Glu in bacterial meningitis was unclear. In this study, we found that meningitis APEC XM infection induced the transport of Ser and Glu from extracellular into intracellular depending on the transporters which activate the mTORC1 pathway, and then induce an inflammatory response in bEnd.3 cells. 

Several studies have investigated the effect of altered AA levels in the CNS during cerebral inflammatory responses and found that changes in AA levels promote the development of various bacterial infections [48,49]. Meningitis bacterial infections can influence AA levels in the CNS, especially those of Glu and Asp, the major excitatory neurotransmitters, which further impair the integrity of the BBB [50,51,52]. Meanwhile, patients with septic encephalopathy present with an imbalance in the levels of branched-chain and aromatic amino acids, resulting in increased levels of IL-6 [53]. In Japanese encephalitis infection or multiple sclerosis patients, Glu content increased significantly and contributed to induce cerebral inflammatory responses, including high production of nitric oxide and superoxide in bEnd.3 cells, resulting in disruption of BBB integrity [51,54,55]. Thus, accumulating pieces of evidence have shown changes in amino acid levels play a critical role during meningitis bacterial infection and neurological diseases. The results of this study showed that APEC XM infection leads to an increase in the intracellular uptake of Ser and Glu in bEnd.3 cells, and a corresponding decrease in the extracellular content of two amino acids (Figure 1). However, there was no process of amino acid uptake at 12 h post-infection, which suggested that amino acid uptake mainly occurred at 1 h post-infection and was involved in APEC XM-colonization and infection in bEnd.3 cells. Moreover, DMEM does not contain alanine and aspartate, so we speculated that their level change may be synthesized by metabolites of carbohydrates or essential amino acids. Since the transport of amino acids depends on amino acid transporters, we demonstrated that the SNAT2 of Ser transporter and EAAT4 of Glu transporter were upregulated during APEC XM infection (Figure 2). Meanwhile, in the Ser/Glu deficient or the 1 mM Ser/1 mM Glu supplement medium (Figure 4 and Figure 5), our study was shown that two amino acids were involved in APEC XM induced upregulation of transporters and inflammation at 1 h [56,57]. These results suggested that increased intracellular levels of Ser and Glu might induce an inflammatory response in a SNAT2 and EAAT4-dependent manner. Further studies are necessary for silencing SNAT2 and EAAT4, as well as overexpressing SNAT2 and EAAT4 to explore amino acid transport depending on transporters. 

We found that SNAT2 silence and EAAT4 silence could be protective against excess inflammation (IL-6, IL-1β, TNF-α, and iNOS) after APEC XM infection (Figure 6 and Figure 7). Additionally, Ser or Glu transport from extracellular into intracellular was also inhibited even when 1 mM Ser or Glu was added. Moreover, compared with the control group, overexpression of SNAT2 or EAAT4 significantly promoted Ser or Glu transport and induced excess inflammation. However, overexpression of SNAT2 or EAAT4 failed to transport Ser or Glu in the Ser or the Glu deficient medium. Our results in cultured bEnd.3 subjected to APEC XM showed that modulating SNAT2 or EAAT4 affected the Ser or Glu transport and pro-inflammatory factors, and it was also proved that the expression of transporters likewise required the participation of amino acids.

Previous reports have demonstrated that signaling pathways are reported to be markedly activated by amino acids for inducing pro-inflammatory factors [58,59]. Recent studies have shown that the mTORC1 pathway has a role in regulating BBB integrity. Inhibition of the mTORC1 pathway can improve cognition in mice, restore microvascular endothelial cell function, and inhibit neurodegeneration in cerebral microvascular endothelial cells [29]. mTORC1 plays an important role in cell growth and activation of the host’s pro-inflammatory factors [60,61]. Normally, the mTORC1 pathway’s activity is mediated through p70S6K, 4E-BP1 and known translational regulators [62], and the mTORC1 pathway has a role in the regulation of physiological and changes [63,64]. Meanwhile, AAs can strongly stimulate mTORC1 activity, thereby maintaining cellular homeostasis and contributing to the regulation of inflammation [28,61]. Thus, we focused on exploring the role of the mTORC1 pathway during APEC XM infection. In our study, we firstly found that the mTORC1 pathway was activated by APEC XM, and then was not activated in the deficiency of Ser and Glu during infection (Figure 4 and Figure 5). These data suggest that the high levels of two intracellular amino acids can activate the mTORC1 pathway, including mTORC1, p70S6K, and 4E-BP1, in bEnd.3 cells. Consistent with the above results, the silence of SNAT2 and EAAT4 could inhibit the activation of the mTORC1 pathway (Figure 6 and Figure 7). The mTORC1 pathway can selectively down-regulate the expression of genes related to amino acid transport in lymphoma cells [65]. Then, we found that mTOR silence could significantly inhibit the expression of SNAT2 and EAAT4, resulting in the inhibition of two amino acid uptakes during APEC XM infection (Figure 8). Simultaneously, the expressions of pro-inflammatory factors were markedly inhibited after infection. These observations have important implications for APEC XM infection of the BBB because activation of the mTORC1 pathway contributes to excess inflammation of BBB. 

Taken together, our findings provide major evidence which firstly demonstrates that the amino acid’s mTORC1 pathway mediates inflammation in bEnd.3 cells during meningitis APEC XM infection. We showed that APEC XM infection induces uptake of Ser and Glu which were transported from extracellular into intracellular, respectively, by increasing the expression of SNAT2 and EAAT4, and then activating the mTORC1 pathway which ultimately leads to the expression of pro-inflammatory factors in bEnd.3 cells.

## 4. Material and Methods

### 4.1. Antibodies and Reagents

DMSO and betaine were obtained from Sigma Aldrich (St. Louis, MO, USA). A cocktail of protease and phosphatase inhibitor cocktails were purchased from MCE (New Jersey, USA). The anti-EAAT4 antibody was obtained from Abcam. The anti-SNAT2 antibody (C-6) was obtained from Santa Cruz Biotechnology. RIPA buffer was purchased from Beyotime (Shanghai, China). Antibodies against phospho-mTOR (Ser2448), mTOR (7C10), phospho-p70S6 Kinase (Thr421/Ser424), phospho-4E-BP1 (Thr37/46), and GAPDH were obtained from CST. All antibodies were used at 1:1000 dilutions. The detailed information of all antibodies is listed in Appendix A.

### 4.2. bEnd.3 Cells, APEC TW-XM Strain Culture and Infection

Mouse brain microvascular endothelial (bEnd.3) cells were cultured in Dulbecco’s modified Eagle medium (DMEM, complete medium) (HyClone Laboratories, Logan, UT, USA) supplemented with 12% fetal bovine serum (FBS) (HyClone Laboratories, Logan, UT, USA). In our study, DMEC is a high sugar type, which mainly contains serine (0.4 mM), threonine, glycine and glutamate (0.4 mM), but does not contain alanine and aspartate. The APEC TW-XM strain (O2:K1:H7) (Ma et al., 2014), belonging to the phylogenetic E. coli reference (ECOR) group B2, was isolated from the brain of infected ducks exhibiting septicemia and neurological symptoms. In this study, we refer to the strain as APEC XM. The APEC XM strain was routinely cultivated in Luria Bertani (LB) broth medium at 37 °C. 

bEnd.3 cells were incubated in DMEM supplemented with 12% FBS at 37 °C with 5% CO_2_ for 24 h. Then, cells were seeded at a density of approximately 5 × 10^5^ cells per 60-mm dish, washed, and incubated in DMEM without or with a multiplicity of infection (MOI) of 100 bacterial cells (APEC XM); monolayers of infected cells were incubated at 37 °C with 5% CO_2_ incubator for 1 h and 12 h. After infection, the samples were stored at −70 °C for further analysis. 

### 4.3. SNAT2, EAAT4 and mTORC1 siRNA Transfection of bEnd.3 Cells

For RNAi experiments, siRNA for *slc38a2*, *slc1a6*, *mTOR*, control was synthesized by GenePharma (Shanghai, China). The primers of the sequences are listed in Appendix A. siRNA (20 μM) for slc38a2, slc1a6, mTOR and was, respectively, transfected into culture bEnd.3 cells for 36 h using siRNA-Mate reagent (GenePharma, Shanghai, China) according to the manufacturer’s instructions. The conditions of cultured bEnd.3 cells were the same as described above. Subsequently, total RNA was extracted to detect the efficiency of the RNAi. Finally, the cells were infected by APEC XM as described above.

### 4.4. Plasmids

We constructed the SNAT2-encoding plasmid (named pEGFP+SNAT2) and the EAAT4-encoding plasmid (named pEGFP+EAAT4), and plasmid pEGFP-C1 (named pEGFP) was used as the control. All plasmid sequences were confirmed by gene sequencing. pEGFP-C1 expression vector was obtained from Invitrogen Co. (Carlsbad, CA, USA).

### 4.5. Amino Acids Analysis

The supernatant was collected 1h post-infection. The cells were added 1 mL of PBS and then treated by repeated freezing and thawing. Finally, the samples were stored at −70 °C for further analysis. Amino acids of cells and the supernatant were analyzed with isotope dilution liquid chromatography–mass spectrometry methods as previously described [66].

### 4.6. RNA Extraction and Quantitative RT-PCR

RNA was extracted and purified from each sample using RNAiso Plus Kit (TIANGEN, Beijing, China) following the manufacturer’s instructions. Purified RNA (2 μg) was used as a template for cDNA synthesis which was performed using FastKing gDNA Dispelling RT SuperMix (TIANGEN, Beijing, China) following the manufacturer’s instructions. The cDNA products were used as a template to quantify the gene expression levels of pro-inflammatory factors (interleukin 1 beta [Il-1β], Il-6, TNF-α and iNOS) and amino acid transporters (Slc1a1, Slc1a2, Slc1a3, Slc1a4, Slc1a5, Slc1a6, Slc1a7, Slc6a14, Slc7a9, Slc7a10, Slc7a11, Slc38a1, Slc38a2, Slc38a4) using the ChamQTM SYBR qPCR Master Mix (Vazyme, Nanjing, China). Real-time qPCR was performed using the ABI 7300 Real-Time PCR System (Applied Biosystems, CA, USA). Primer sequences are illustrated in Appendix A. The expression levels of pro-inflammatory and amino acid transporter-related genes were normalized to that of glyceraldehyde-3-phosphate dehydrogenase (GADPH; internal control). 

### 4.7. Western Blot Analysis 

Treated bEnd.3 cells were washed twice with ice cold PBS and lysed in RIPA buffer containing phenylmethylsulfonyl fluoride (PMSF), as described elsewhere (Zhu et al., 2016). The cell debris was pelleted by centrifugation at 13,000 rpm for 10 min at 4 °C and proteins in the cell lysates were separated by 12% SDS PAGE and transferred to polyvinylidene fluoride (PVDF) membranes (Millipore, MA, USA). The membranes were blocked in 5% dry milk in TBST for 1 h at room temperature, followed by incubation with primary antibodies (1:1000) over night at 4 °C, and subsequently with HRP conjugated secondary antibodies (1:5000) for 1 h at room temperature. The following antibodies were used: anti-SNAT2, anti-EAAT4, anti-mTOR, anti-phospho-mTOR, anti-phospho-p70S6K, anti-phospho-4E-BP, and anti-GAPDH. Protein bands were visualized by chemiluminescence methods.

### 4.8. Statistical Analysis

Differences between means were tested by ANOVA. If the ANOVA rejected the null hypothesis of the same means among the conditions (*p* < 0.01), comparisons were performed between selected pairs of means by two-tailed unpaired *t*-test (* *p* < 0.05, ** *p* < 0.01) using GraphPad Prism version 6 (GraphPad Software, La Jolla, CA, USA). At least three biological replicates were used in each experiment.

## Figures and Tables

**Figure 1 ijms-22-09245-f001:**
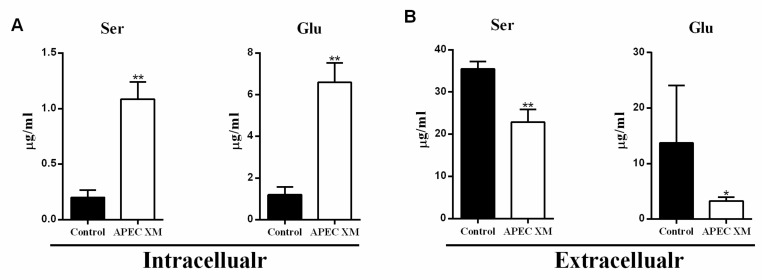
APEC XM infection promotes uptake of serine, and glutamate in bEnd.3 cells at 1 h. (**A**) The intracellular content of serine and glutamate; (**B**) The extracellular content of serine and glutamate. Data are from one experiment representative of three independent experiments, with six replicates per group. Data represent means ±SD. * *p* < 0.05, ** *p* < 0.01.

**Figure 2 ijms-22-09245-f002:**
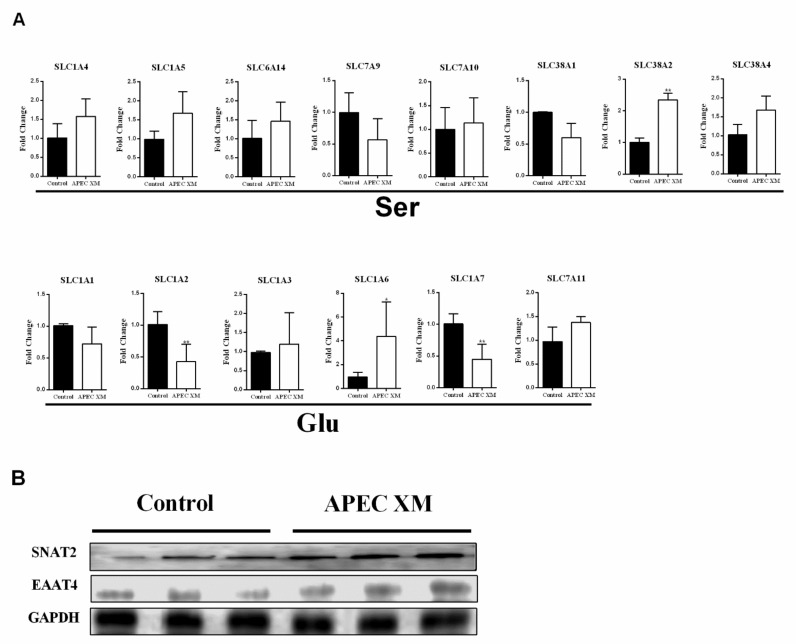
APEC XM infection upregulates the expression of SNAT2 and EAAT4. (**A**) RT-PCR validation of the expression of *slc1a1*, *slc1a2*, *slc1a3*, *slc1a4*, *slc1a5*, *slc1a6*, *slc1a7*, *slc6a14*, *slc7a9*, *slc7a10*, *slc7a11*, *slc38a1*, *slc38a2* and *slc38a4* in bEns.3 cells at 1 h post-infection. *GAPDH* was used as the reference control. Data are from one experiment representative of three independent experiments, with six replicates per group. Data represent means ± SD. * *p* < 0.05, ** *p* < 0.01. (**B**) Western blotting showed the significant upregulation of SNAT2 and EAAT4 in bEnd.3 cells at 1 h post-infection. The *GAPDH* was used as the loading control.

**Figure 3 ijms-22-09245-f003:**
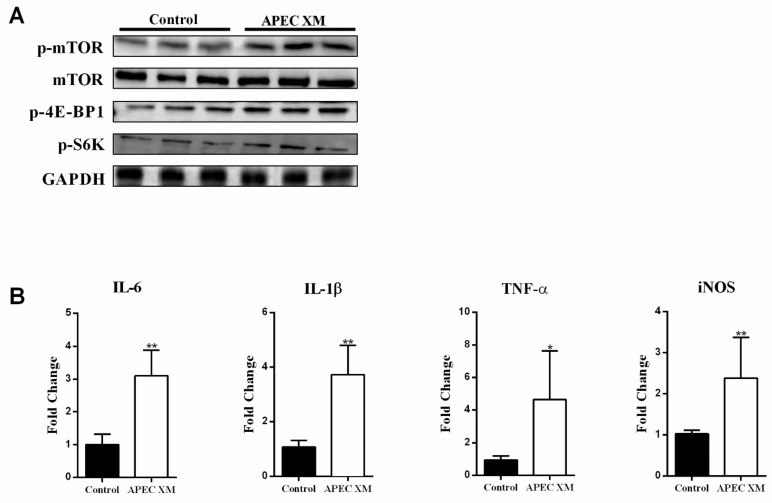
APEC XM activates the mTORC1 pathway and induces inflammation at 1 h post-infection. (**A**) Western blotting showed that up-regulation of phospho-mTOR, mTOR, phospho-4E-BP1 and phosphor-p70 S6 in bEnd.3 cells at 1 h post-infection. The *GAPDH* was used as the loading control. (**B**) RT-PCR validation the mRNA levels of pro-inflammatory factors after APEC XM infection. Data are from one experiment representative of three independent experiments, with three to six replicates per group. Data represent means ± SD. ns: not significant, * *p* < 0.05, ** *p* < 0.01.

**Figure 4 ijms-22-09245-f004:**
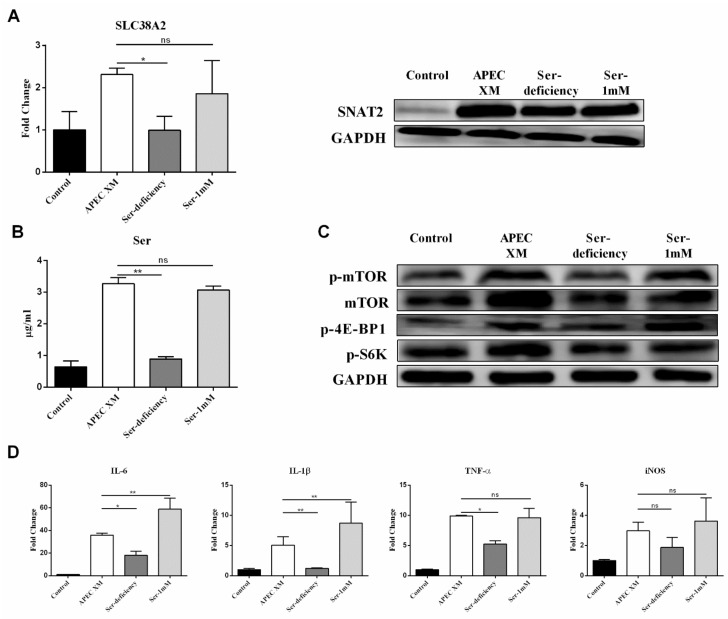
Serine involved in APEC XM-induced the expression of SNAT2, activation of mTOR pathway and inflammation in bEnd.3 Cells. (**A**) The transcription and expression levels of Slc38a2 in bEnd.3 cells in the Ser deficiency or the Ser supplement medium compared with APEC XM infection cells by RT-PCR and western blot analyses. *GAPDH* and GAPDH were used as the reference control for RT-PCR and the blotting. The RT-PCR data were from one experiment representative of three independent experiments and were presented as mean ± SD. (**B**) The intracellular content of Ser in bEnd.3 cells in the Ser deficiency or the Ser supplement medium was respectively significantly decreased and no significant difference compared with APEC XM infection cells. (**C**) Western blotting showed the significant activation of the mTORC1 pathway in bEnd.3 cells in the Ser deficiency or the Ser supplement medium. GAPDH was used as the reference control. (**D**) The mRNA expressions of pro-inflammatory factors in bEnd.3 cells in the Ser deficiency or the Ser supplement medium compared with APEC XM-infected cells were shown. *GAPDH* was used as the reference control. Data are from one experiment representative of three independent experiments, with three to six replicates per group. Data represent means ± SD. ns: not significant, * *p* < 0.05, ** *p* < 0.01.

**Figure 5 ijms-22-09245-f005:**
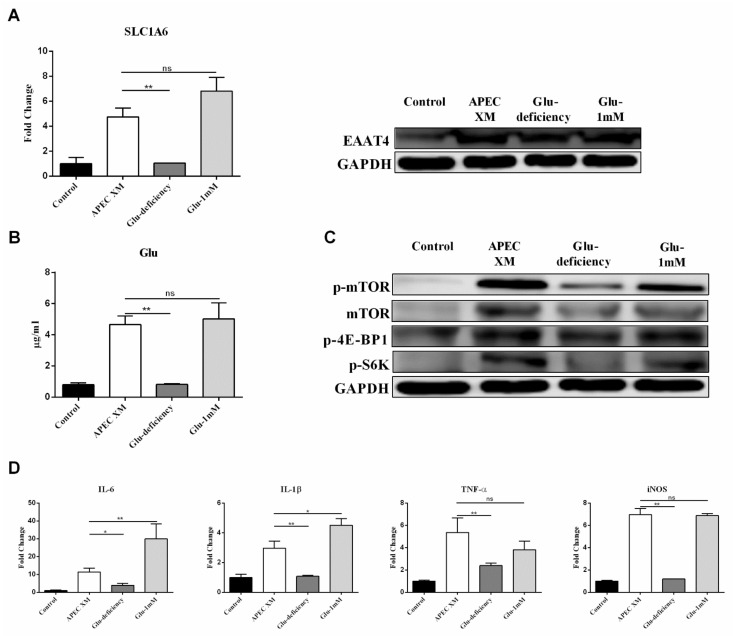
Glutamate involved in APEC XM-induced the expression of EAAT4, activation of mTOR pathway and inflammation in bEnd.3 Cells. (**A**) The transcription and expression levels of Slc1a6 in bEnd.3 cells in the Glu deficiency or the 1 mM Glu supplement medium compared with APEC XM infection cells by RT-PCR and western blot analyses. *GAPDH* and GAPDH were used as the reference control for RT-PCR and the blotting. The RT-PCR data were from one experiment representative of three independent experiments and were presented as mean ± SD. (**B**) The intracellular content of Glu in bEnd.3 cells in the Glu deficiency or the 1 mM Glu supplement medium was respectively significantly decreased and no significant difference compared with APEC XM infection cells. The data were from one experiment representative of three independent experiments and were presented as mean ± SD. (**C**) Western blotting showed the significant activation of the mTORC1 pathway in bEnd.3 cells in the Glu deficiency or the 1 mM Glu supplement medium. GAPDH was used as the reference control. (**D**) The mRNA expressions of pro-inflammatory factors in bEnd.3 cells in the Glu deficiency or the 1 mM Glu supplement medium compared with APEC XM-infected cells were shown. *GAPDH* was used as the reference control. Data are from one experiment representative of three independent experiments, with three to six replicates per group. Data represent means ± SD. ns: not significant, * *p* < 0.05, ** *p* < 0.01.

**Figure 6 ijms-22-09245-f006:**
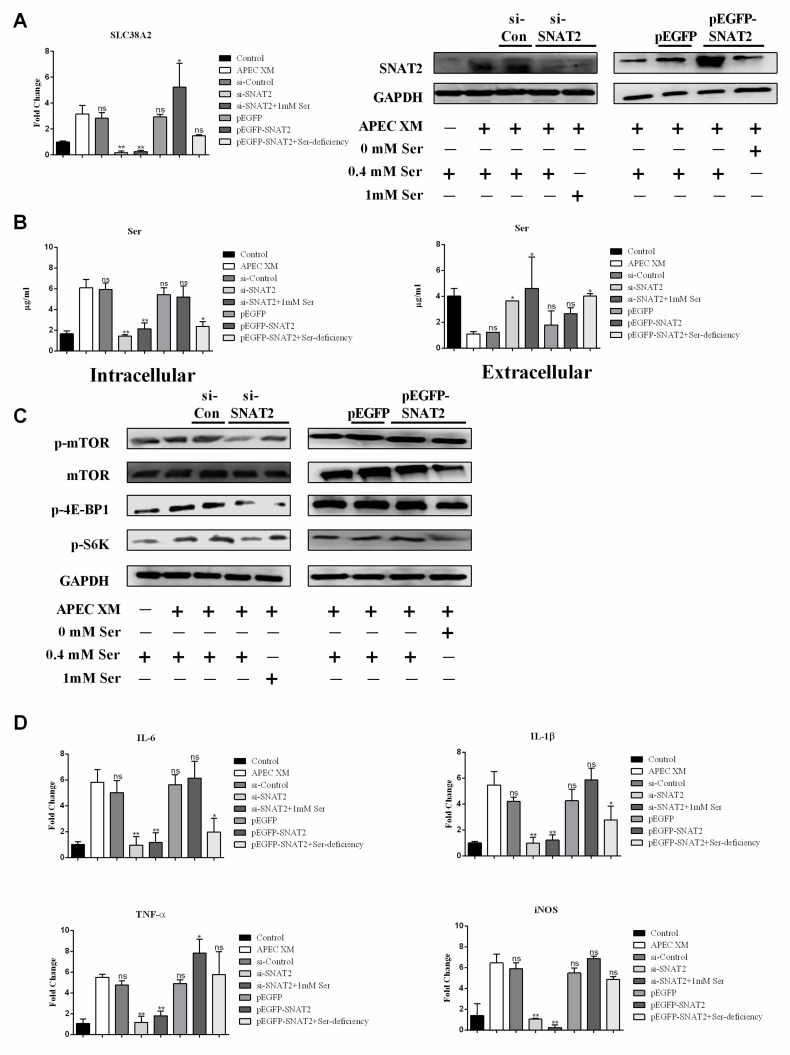
The effect of SNAT2 silencing and overexpression on Ser uptake, activation of the mTORC1 pathway, and inflammation. (**A**) The mRNA transcription and expression of *Scl38a2*, respectively in SNAT2-silenced cells cultured in the complete or the 1 mM Ser supplement medium, or in SNAT2-overexpressed cells cultured in the complete or the Ser deficiency medium, compared with APEC XM-infected cells. *GAPDH* and GAPDH were used as the reference control for RT-PCR and the blotting. Data are from one experiment representative of three independent experiments, with three to six replicates per group. Data represent means ± SD. (**B**) The intracellular and extracellular content of Ser, respectively in SNAT2-silenced cells cultured in the complete or 1 mM the Ser supplement medium, or in SNAT2-overexpressed cells cultured in the complete or the Ser deficiency medium, compared with APEC XM-infected cells. Data are from one experiment representative of three independent experiments, with three to six replicates per group. Data represent means ± SD. (**C**) Western blotting showed the significant activation of the mTORC1 pathway, respectively in SNAT2-silenced cells cultured in the complete or the 1 mM Ser supplement medium, or in SNAT2-overexpressed cells cultured in the complete or the Ser deficiency medium, compared with APEC XM-infected cells. GAPDH was used as the reference control for the blotting. (**D**) The mRNA transcription of pro-inflammatory factors, respectively in SNAT2-silenced cells cultured in the complete or the 1 mM Ser supplement medium, or in SNAT2-overexpressed cells cultured in the complete or the Ser deficiency medium, compared with APEC XM-infected cells. *GAPDH* was used as the reference control for RT-PCR. Data are from one experiment representative of three independent experiments, with three to six replicates per group. Data represent means ± SD. ns: not significant, * *p* < 0.05, ** *p* < 0.01.

**Figure 7 ijms-22-09245-f007:**
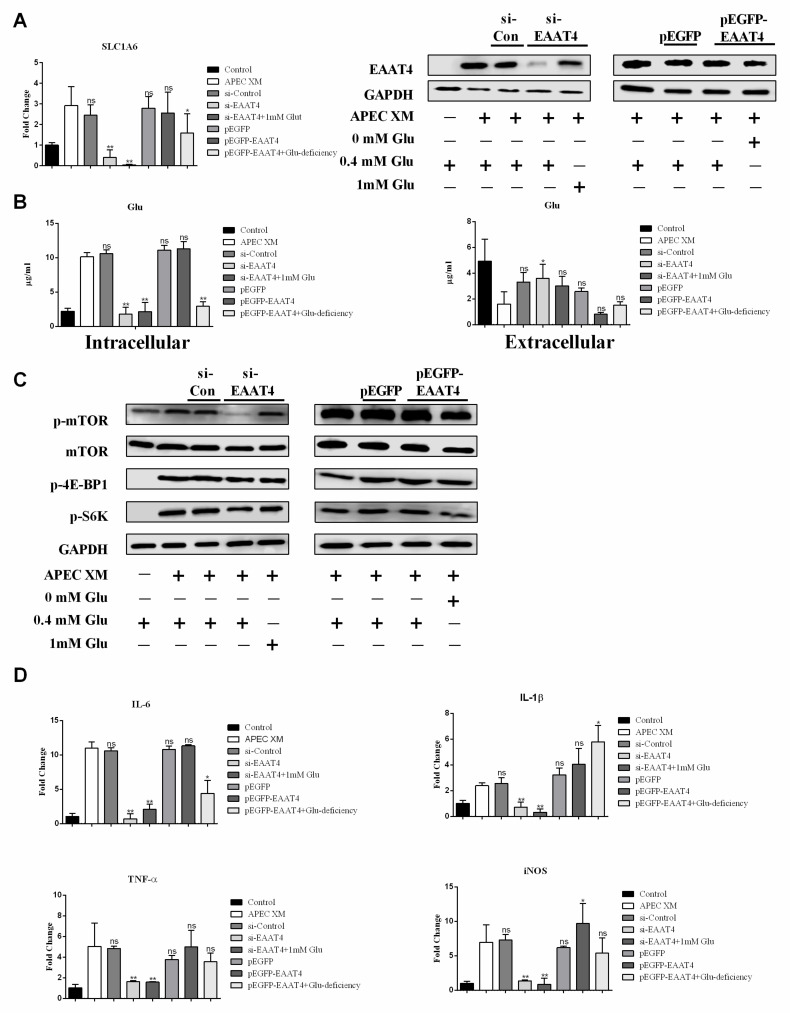
The effect of EAAT4 silencing and overexpression on Glu uptake, activation of the mTORC1 pathway, and inflammation. (**A**) The mRNA transcription and expression of *Scl1a6*, respectively in EAAT4-silenced cells cultured in the complete or the 1 mM Glu supplement medium, or in EAAT4-overexpressed cells cultured in the complete, or the Glu deficiency medium, compared with APEC XM-infected cells. *GAPDH* and GAPDH were used as the reference control for RT-PCR and the blotting. Data are from one experiment representative of three independent experiments, with three to six replicates per group. Data represent means ± SD. (**B**) The intracellular and extracellular content of Glu, respectively in EAAT4-silenced cells cultured in the Glu or the 1 mM Glu supplement medium, or in EAAT4-overexpressed cells cultured in the complete or the Glu deficiency medium, compared with APEC XM-infected cells. Data are from one experiment representative of three independent experiments, with three to six replicates per group. Data represent means ± SD. (**C**) Western blotting showed the significant activation of the mTORC1 pathway, respectively in EAAT4-silenced cells cultured in the complete or the 1 mM Glu supplement medium, or in EAAT4-overexpressed cells cultured in the complete or the Glu deficiency medium, compared with APEC XM-infected cells. GAPDH was used as the reference control for the blotting. (**D**) The mRNA transcription of pro-inflammatory factors, respectively in EAAT4-silenced cells cultured in the complete or the 1 mM Glu supplement medium, or in EAAT4-overexpressed cells cultured in the complete or the Glu deficiency medium, compared with APEC XM-infected cells. *GAPDH* was used as the reference control for RT-PCR. Data are from one experiment representative of three independent experiments, with three to six replicates per group. Data represent means ± SD. ns: not significant, * *p* < 0.05, ** *p* < 0.01.

**Figure 8 ijms-22-09245-f008:**
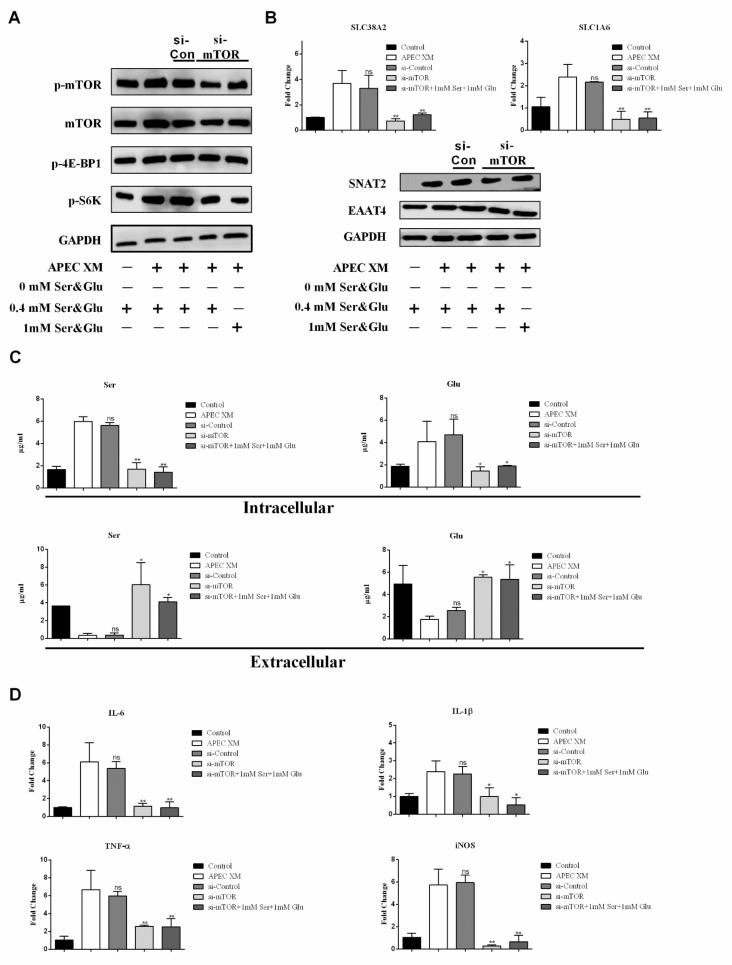
The effect of mTOR silencing on Ser or Glu uptake, expression of SNAT2 or EAAT4, activation of mTORC1 pathway and inflammation. (**A**) Western blotting showed the downregulated activation of the mTORC1 pathway, respectively in mTOR-silenced cells cultured in the compete, the 1 mM Ser supplement or the 1 mM Glu supplement medium, compared with APEC XM-infected cells. GAPDH was used as the reference control for the blotting. (**B**) The mRNA transcription and expression of *Scl38a2* and *Scl1a6*, respectively in mTOR-silenced cells cultured in the complete, the 1 mM Ser supplement or the 1 mM Glu supplement medium, compared with APEC XM-infected cells. *GAPDH* and GAPDH were used as the reference control for RT-PCR and the blotting. RT-PCR data are from one experiment representative of three independent experiments, with three to six replicates per group. Data represent means ± SD. ns: not significant, * *p* < 0.05, ** *p* < 0.01. (**C**) The intracellular and extracellular content of Ser and Glu, respectively in mTOR-silenced cells cultured in the complete, the 1 mM Ser supplement or the 1 mM Glu supplement medium, compared with APEC XM-infected cells. Data are from one experiment representative of three independent experiments, with three to six replicates per group. Data represent means ± SD. ns: not significant, * *p* < 0.05, ** *p* < 0.01. (**D**) The mRNA transcription of pro-inflammatory factors, respectively in mTOR-silenced cells cultured in the complete, the 1 mM Ser supplement or the 1 mM Glu supplement medium, compared with APEC XM-infected cells. *GAPDH* was used as the reference control for RT-PCR. Data are from one experiment representative of three independent experiments, with three to six replicates per group. Data represent means ± SD. ns: not significant, * *p* < 0.05, ** *p* < 0.01.

## Data Availability

All datasets generated for this study are included in the article/Appendix A.

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
