# Peer review of "The Amino Acid-mTORC1 Pathway Mediates APEC TW-XM-Induced Inflammation in bEnd.3 Cells"

_ijms, 2021, doi:10.3390/ijms22179245_

Round 1

Reviewer 1 Report

This study seeks to determine how APEC TW-XM-stimulated Ser or Glu uptake activates inflammation in bEnd.3 cells. The authors examined the Ser and Glu transporter expression, some of mTORC1 and pro-inflammatory pathway mediators.

Overall, the Western blots are of a relatively poor quality and saturated control protein levels in many samples, in addition, both overexpression and knockdown data are weak. Therefore, it doesn’t seem enough to support the story. 

Moreover, the authors do not provide solid evidences that mTORC1 is a key mediator of inflammation induced by AA uptake in bEnd.3 cells. For instance, mTORC1 upstream or mTORC2 component can be also involved in this mechanistic pathway. It would be helpful if this information is included.

Author Response

  1. Overall, the Western blots are of a relatively poor quality and saturated control protein levels in many samples, in addition, both overexpression and knockdown data are weak. Therefore, it doesn’t seem enough to support the story.

Thanks for your comments. We may think In the WB results are of a relatively poor quality with many potential unknown reasons , therefore, we try to adjust the size of some of the result images, such as fig4C-p-4E-BP1; at the same time, we re-detected some of the poor quality results, such as Fig 2B-SNAT2, fig3A-p-4E-BP1, fig7C-p-mTOR, fig8B-SNAT2; and the results of each protein in the revised manuscript showed that the protein expression of the control group was significantly lower than that of the APEC TW-XM infection group. After silencing transporters and mTOR proteins, firstly, at the RNA and protein levels, compared with the control group, the expression of each silencing protein was significantly reduced. The silencing of the three proteins also has a significant effect on the expression of the target detection protein compared with APEC XM-infection group. The overexpression of the two transporters has no significant difference in the effect of the target detection protein compared with APEC XM-infection group, but the activation of mTORC1 pathway is higher than that of the silence and control group. We adjusted the description of results section, respectively line 235~240, with highlighted in red.

  1. Moreover, the authors do not provide solid evidences that mTORC1 is a key mediator of inflammation induced by AA uptake in bEnd. 3 cells. For instance, mTORC1 upstream or mTORC2 component can be also involved in this mechanistic pathway. It would be helpful if this information is included.

Thanks for your comments. We should highlight the mTORC1 background in the early version. In the revised one, we have added the contents of mTORC2 in introduction section, aimed to explain why we just only detected the role of mTORC1 in regulating of inflammation induced by AA uptake in bEnd.3 cells during infection. In the revised version, I have added more explanations/descriptions to the bacterial infection impact of this work with supplementary contents highlighted in red, such as line 68~77.

Reviewer 2 Report

In this research article the authors checked a new pathways, Amino acid-mTORC1 in vitro in the brain endothelial cells. Overall the data is well presented, and the experiments are conducted carefully, however the data needs a little improvements. In the manuscript no figure number is mentioned so it is difficult for the readers to understand. The others comments are listed below accordingly.

  1. Abbreviation explained in the beginning of every section e.g., in the beginning of results section
  2. Why the cells were infected for 1 and 12 hours respectively?
  3. Each figure needs a clear figure legends
  4. In figure 2b western blot SNAT2, in control and in infected cells no significant changes observed, it needs to conduct again
  5. In figure 3a. p-4E-BP1, is not looking clear, it needs improvements
  6. Figure 4 is looking unarranged, it needs arrangement,
  7. All figure 5 also needs arrangements
  8. Mention the number for each figure in manuscript
  9. Figure numbers are not mentioned ,it is difficult to understand
  10. Last figure needs arrangements
  11. Discussion needs improvements
  12. In discussion section all results should explained briefly, according to figure numbers
  13. A clear conclusion also include in the manuscript
  14. Make a table for all antibody, their source, catalog number etc. mention in the table

Author Response

  1. Abbreviation explained in the beginning of every section e.g., in the beginning of results

Thanks for your good suggestion. We have added explanations of all the abbreviations with supplementary contents highlighted in red. mTORC1 (mechanistic target of rapamycin complex 1) in line 23. IL-1β (interleukin-1beta) in line 60. APEC TW-XM (avian pathogenic Escherichia coli TW-XM) in line 84. SNAT2, sodium-dependent neutral amino acid transporter 2) in line121. EAAT4, excitatory amino acid transporter 4 in line 125.

  1. Why the cells were infected for 1 and 12 hours respectively?

Thanks for your good comments. The infectious process of APEC XM, we previously found that bacterial infection for 1h was early infection and bacterial infection for 12h infection was late infection. Bacterial infection for 1h was the process of bacterial adhesion and invasion, which was the key step during infection. And bacterial infection for 12h was the process of inducing bEnd.3 cells apoptosis, which was another key step during infection. In addition, bacterial infection could induce cells metabolic disorder and amino acids were reported that involved in two processes. Thus, we would like chose two time point that the cells were infected for 1 and 12 hours, respectively.

  1. Each figure needs a clear figure legends

Thanks for your good comments. We try to reorganize the figure legends to be understood well which listed in the end of manuscript with changes highlighted in red.

  1. In figure 2b western blot SNAT2, in control and in infected cells no significant changes observed, it needs to conduct again

Thanks for your good suggestion. In order to understand clearly, the expression of SNAT2 in figure 2B was retested and we remake it again.

  1. In figure 3a. p-4E-BP1, is not looking clear, it needs improvements

Thanks for your good suggestion. The expression of p-4E-BP1 in figure 3A was retested and we remake it again.

  1. Figure 4 is looking unarranged, it needs arrangement,

Thanks for your good suggestion. We have rearranged figure 4 to be understood better.

  1. All figure 5 also needs arrangements

Thanks for your good suggestion. We have rearranged figure 5 to be understood better.

  1. Mention the number for each figure in manuscript

Thanks for your good suggestion. We have added the number for each figure in manuscript with supplementary number highlighted in red.

  1. Figure numbers are not mentioned, it is difficult to understand

Thanks for your good suggestion. We have added the number for each figure in manuscript with supplementary figure numbers highlighted in red.

  1. Last figure needs arrangements

Thanks for your good suggestion. We have rearranged figure 8 to be understood better.

  1. Discussion needs improvements

Thanks for your good suggestion. We try to improve discussion with changes highlighted in red, line 320~322,349~350,354~355.

  1. In discussion section all results should explained briefly, according to figure numbers

Thanks for your good suggestion. We have tried to explain all results briefly according to figure numbers in discussion section with changes highlighted in red.

  1. A clear conclusion also include in the manuscript

Thanks for your good suggestion. We made a clear conclusion in manuscript which highlighted in red, line 370~373.

  1. Make a table for all antibody, their source, catalog number etc. mention in the table

Thanks for your good suggestions. We added it, and have listed the detailed information of all antibodies in a table, named as Supplementary Table S3 in supplementary material.

Round 2

Reviewer 1 Report

The authors have addressed my concerns. Thank you.